# Facile Preparation of Flame Retardant Cotton Fabric via Adhesion of Mg(OH)_2_ by the Assistance of Ionic Liquid

**DOI:** 10.3390/polym12020259

**Published:** 2020-01-22

**Authors:** Jinli Ma, Xiao Wang, Jing Li, Ru Chen, Ju Wei

**Affiliations:** 1School of Textile and Material Engineering, Dalian Polytechnic University, Dalian 116034, China; majili0718@163.com (J.M.); chenru2003@163.com (R.C.); weiju@dlpu.edu.cn (J.W.); 2Argus (Shanghai) Enterprise Group, Shanghai 201800, China; anjing430X@163.com

**Keywords:** Mg(OH)_2_, ionic liquid, cotton fabric, physical adhesion, flame retardancy

## Abstract

A new approach for flame retardant functional finishing of textiles was explored to improve flame retardancy of cotton fabrics by simple physical adhesion method. Mg(OH)_2_ was adhered to cotton fiber with the aid of fiber swelling in ionic liquid on heating and shrinkage on washing to obtain flame retardancy. The effects of immobilizing condition and methods on flame retardancy were discussed. The surface morphology, crystal structure, combustion behavior, thermal and physical properties of cotton fabric adhered with Mg(OH)_2_ were analyzed. The afterflame time and afterglow time of adhered cotton fabric were significantly reduced to less than 5 s. The thermal weight loss of cotton fabric was increased by 11.7% and the total heat released per unit mass was decreased by 20.9% after MH adhesion. The simple eco-friendly adhesion method provided a convenient approach for the development and application of flame retardant functional cellulosic textiles.

## 1. Introduction

Cotton is widely used in apparel fabrics, decorative fabrics, and industrial fabrics because of its excellent moisture absorption, breath resistance, alkali resistance, heat insulation, spinnability, and biodegradability [1]. But the flammability of cotton fibers limits their widespread use, especially in functional and smart textiles [2,3,4]. At present, various methods have been used to impart flame retardant properties to cotton fibers, for example, solution-dipping [5], grafting [6], sol-gel [7], layer-by-layer assembly technique [8], etc. Most of the flame retardants currently used for cotton fabrics via this method are organic flame retardants containing elements such as phosphorus, nitrogen and silicon [9,10,11,12]. These organic flame retardants have improved the flame retardancy of cotton fabrics. Nevertheless, considering the potential neurotoxicity and carcinogenicity of some organic flame retardants [13,14], the ecological safety and environmentally friendliness of flame retardants are more concerned nowadays.

In order to develop environmentally friendly flame retardants while maintaining better fire performance, inorganic compounds, such as metal hydroxides, are of increasing interest. Mg(OH)_2_ (MH) has attracted more and more attention as an inorganic flame retardant because of its multiple flame retardant effects, such as low smoke and non-toxicity. Mao Z et al. had assembled lamellar-like and rod-like MH crystals on the surface of cotton fibers by treatment with urea and citric acid solution to reduce the thermal weight loss of cotton when it was heating [15]. Han Y et al. utilized wasted cotton fabrics to make flame retardant cellulose aerogel by in-situ synthesis of MH nanoparticles in a cellulose gel nanostructure. The aerogel exhibits excellent flame retardancy, which can be extinguished in 40 s [16]. Wang X et al. introduced vinyl-containing silane-modified MH hybrid pigment particles onto the surface of cellulose fibers via surface-initiated ATRP grafting. The afterflame time and afterglow time of treated fabric were 2.8 and 16.1 s, respectively, comparing with 4.3 and 31.7 s of cotton fabric [17]. Although flame retardancy of cellulose is obtained via MH introduction, complex chemical processes are involved. It is a new trial is to physically immobilize MH on the surface of fiber under swelling without modification of MH and cellulose fiber. There are many researches on the use of ionic liquid (IL) to dissolve and regenerate cellulose [18,19,20,21,22]. However, there are relatively few studies about using IL to swell cellulose for immobilizing inorganic particles to achieve functional fabric.

In this research, a new method for flame retardant functional finishing of textiles is explored, which can improve the flame retardancy of cotton fabric (CF) by facile physical adhesion method for the immobilization of MH with the aid of fiber swelling in IL and shrinkage in water. Adhesion of MH will occur on the surface of cotton fibers after swelling and partial dissolving by IL of 1-allyl-3-methylimidazolium chloride ([AMIM]Cl) via heating or heating under pressure. The adhesion of MH will be realized by shrinkage of cotton fibers in water. The surface morphology and crystal structure of CF adhered with MH (CF/MH) is analyzed. In addition, the flame retardancy, thermal stability, and combustion behavior of CF/MH are also investigated. This method is hoped to improve the flame retardant properties of CF, and explore a new way for flame retardant functional of textile finishing.

## 2. Experimental

### 2.1. Synthesis and Purification of IL and MH/IL Suspension

1-methylimidazole (Beijing Belling Technology Co., Ltd., Beijing, China) and allyl chloride (Sigma Aldrich Co., Ltd., Shanghai, China) were mixed at a ratio of 1:1.2. The mixture was stirred and heated to 60 °C under a N_2_ atmosphere. After 7 h of reaction, a crude product of [AMIM]Cl was obtained. After rotary evaporation (RE-52C, Yuhua Instrument Co., Ltd., Gongyi, China) was completed, the crude product was cooled down to room temperature and extracted with diethyl ether (Tianjin Kermel Chemical Reagent Co., Ltd., Tianjin, China). After drying in vacuum oven (VD-216, Done-E Industry Co., Ltd., Shanghai, China) at 80 °C for 48 h, a pale yellow viscous [AMIM]Cl IL was obtained [23]. As for in situ synthesis of MH in IL, the MH powder (99%, Beijing DK Nano Technology Co., Ltd., Beijing, China) was slowly added into 1-methylimidazole. After stirring and ultrasonically dispersing (Probe type ultrasonic disperserFS-600, Shanghai Senxin Experimental Instrument Co., Ltd., Shanghai, China), the suspension was mixed with allylchloride to prepare MH/IL suspension via in situ synthesis. The methods of synthesis and purification were same as above.

### 2.2. Swelling of CF in IL

The cotton fiber, after dipping in IL (1 mL), was heated to 80 °C at a heating rate of 20 s/°C and photographed at a fixed time interval using a polarizing hot stage microscope (Shanghai Hui Tong Optical Instrument Co., Ltd.) to observe swelling and dissolution. CF (2 cm × 2 cm) was placed in a beaker containing IL, heated to 80 °C, and then kept at a constant temperature. The swelling of CF was observed at intervals.

### 2.3. Spray of MH/IL Suspension and Adhesion of MH onto CF

Two methods of hot air and hot stamping were used for adhesion of MH onto CF. Two edges of CF (36 cm × 11 cm) in the warp direction were fixed on the needle plate holder, which was placed on a conveyor belt running at 0.5 m/s. The MH suspension was sprayed evenly on the double surface of CF under the conditions of which MH amount was 5 wt% and spraying time was 3. Then, two ends of the sprayed CF were fixed on the needle plate and heated under tension in electric thermostatic drying oven (STIK (Shanghai) Co., Ltd., Shanghai, China). CF/MH was washed to remove the IL. After the fabric was dried, CF/MH via hot air was obtained. The two ends of the sprayed CF were fixed on the needle plate and dried under vacuum at room temperature for 2 h. Hot stamping of the sprayed fabric was treated with a high pressure shaking head ironing machine (HF3805, Jinan Huafei Information Technology Co., Ltd., Jinan, China). Then, it was washed and dried to obtain a CF/MH via hot stamping. The mass, warp dimension, and weft dimension of CF before and after adhesion of MH onto fabric were measured. The adhesion ratio of MH onto CF and the shrinkage of CF/MH after washing were calculated by the following formula:$$\mathrm{Adhesion}\;\mathrm{ratio}\;\mathrm{of}\;\mathrm{MH}\;=\;\frac{m-m_{0}}{m_{0}}\times 100\%$$
m_0_ and m represent the mass of CF before and after MH adhesion.
$$\mathrm{Warp}\;\mathrm{shrinkage}\;\mathrm{ratio}\;=\;\frac{j-j_{0}}{j_{0}}\times 100\%$$
j_0_ and j represent warp length of CF before and after MH adhesion.
$$\mathrm{Weft}\;\mathrm{shrinkage}\;\mathrm{ratio}\;=\;\frac{W-W_{0}}{W_{0}}\times 100\%$$
w_0_ and w represent weft length of CF before and after MH adhesion.

### 2.4. Characterization and Analysis

Surface morphology of CF and CF/MH was observed by SEM (JSM-6460LV, JEOL, Tokyo, Japan). The crystal structure was analyzed by X-Ray Diffractomer (D/max-3BX, Rigaku, Japan) with a scanning range of 5°–65° and a scanning speed of 5°/min. The chemical structure were characterized using Fourier transform infrared spectra (Spectrum One-B, PerkinElmer, Waltham, MA, USA). The afterflame time and afterglow time were measured using YG815 vertical burner (Shandong Anqiu Jingwei Textile Instrument Co., Ltd., Anqiu, China) according to GBT 5455-2014. The samples for the following analysis are CF/MH with adhesion ratio of MH of 7.99%, warp and weft shrinkage of 1.09% and 15.24% via hot air, and 7.36%, 1.40%, and 2.13% via hot stamping. Thermogravimetric analysis (TGA) and differential thermal gravity (DTG) were analyzed by a Differential Thermal Analyzer (STA PT1600, Lineis, Germany) from 20 °C to 500 °C in N_2_ atmosphere with a heating rate of 10 °C/s. The combustion behavior was analyzed by the cone calorimetry (Vouch 6810, Suzhou, China) under an external heat flux of 50 kW/m^2^ according to ISO 5660-1. The flexibility and activity ratio were measured on the fabric style tester (YG821L, Laizhou Electronic Instrument Co., Ltd., Yantai, China) according to FZ/T01054. According to FZ/T80007, the breaking strength was measured using YG065 electronic fabric strength tester (Laizhou Electronic Instrument Co., Ltd., Yantai, China).

## 3. Results and Discussion

### 3.1. CF Swelling in IL

It can be seen from Figure 1, the cotton fiber becomes brighter and thicker due to swelling by IL when heated for 1 min at 80 °C. Meanwhile, natural convolution of cotton fiber gradually disappears, indistinct fiber boundary is noticed. No significant change in the fiber diameter was further observed with prolonged time due to limited amount of IL. As shown in Figure 2, obvious swelling phenomenon is firstly noticed for the yarns at the edge of CF and the structure of CF is visible when the CF is heated at 80 °C for 1 min in excessive IL. Nevertheless, CF shrinks largely when the time reaches 2 min. The shrinkage of CF is attributed to anisotropic swelling of cotton fibers in radical and lengthwise direction. A gel state of CF is found due to partial dissolution of cellulose by IL after 3 min. The above phenomena indicate that controllable swelling of cotton fibers can be obtained when adjusting IL amount and heating temperature. Moreover, obvious swelling effect of CF by IL can be achieved with limited amount of IL for several minutes above 80 °C while maintaining the shape of CF, providing a reference to design reasonable process of MH immobilization. The adhesion of MH will adhere to surface cellulose fibrils after coagulation during washing of CF in water.

### 3.2. Effect of Adhesion Conditions on Flame Retardancy of CF

It can be seen from Table 1 the increment of spraying temperature and heating time leads to higher adhesion ratio of MH to CF. The swelling anisotropy and internal stress relaxation of the fiber cause the fabric to shrink in the warp direction and weft direction. The weft shrinkage and warp shrinkage are mainly due to heating time, which means the swelling of cotton fiber by IL. Higher weft shrinkage than warp shrinkage result from no tension in the weft direction during treatment. The higher spraying temperature results in more sprayed MH/IL suspension at a lower viscosity, with a resultant higher adhesion ratio of MH under more severe swelling of cotton fibers. The afterflame time and afterglow time are reduced with the adhesion ratio of MH increasing. Nevertheless, weft and warp shrinkage prolong the afterflame time and afterglow time at a higher adhesion ratio of MH, respectively, since thicker and more compact CF can lower the flame spread rate and greater afterflame time, as well as a reduction in afterglow time.

### 3.3. Effect of Heating Methods on Flame Retardancy of CF

It can be seen from Table 2 the adhesion ratio of MH is enhanced with the heating temperature increasing for two adhesion methods due to improved swelling and shrinkage effect of IL on the surface fibrils of the cotton fiber at higher temperature, resulting in higher adhesion ratio of MH. The weft shrinkage of CF adhered with MH via hot stamping is lower than hot air. Hot stamping limits the shrinkage of the fabric due to pressure. When the MH adhesion ratio exceeds 6.28%, the afterflame time and the afterglow time of CF can be kept within 5 s via hot stamping.

### 3.4. SEM Analysis

It can be seen from Figure 3a that the surface of the CF is relatively smooth. Wrinkles appear on the surface of the cotton fiber swelled by IL and the fibers are relatively thicker due to the swelling effect of IL (Figure 3b,c). MH particles are observed to be entrapped on the surface of the cotton fiber swelled by IL (Figure 3d). Since cellulosic fibrils on the fiber surface re-solidify after being swollen and partially dissolved by IL, adhesion occurs between some of the fibers swelled by IL via hot air and hot stamping (Figure 3e,f). MH particles are not easy to be seen on the fiber surface due to formation of certain cellulosic film by stronger swelling, partial dissolution, and resolidification via hot air than hot stamping (Figure 3g,h), which is consistent with larger shrinkage of CF via hot air in Table 1 and Table 2. However, the roughness of the CF is increased and MH particles are obviously observed on the fiber surface via hot stamping(Figure 3h), which is conducive to protection of CF for a shorter afterflame time and afterglow time as shown in Table 2.

### 3.5. XRD Analysis

The MH diffraction pattern in Figure 4 shows seven diffraction peaks at 2θ of 18.66°, 33.10°, 38.25°, 50.96°, 58.90°, 62.36° and 68.39°, which correspond to the diffraction from (001) crystal plane, (100) crystal plane, (101) crystal plane, (102) crystal plane, (110) crystal plane, (111) crystal plane, and (103) crystal plane. Four diffraction peaks of CF appear at 2θ of 14.85°, 16.49°, 22.76°, and 34.32°, which are attributed to the diffraction of crystal plane (101), ($10\overset{-}{1}$), (002), and (040) of cellulose I [24,25]. All the diffraction peaks of cotton fiber cellulose I and MH appear in the XRD curve of the CF/MH, indicating that MH is adhered to CF. In addition, compared to CF, the XRD curve of the CF swelled via hot air and CF/MH via hot air show a diffraction with weak peak at the 2θ of 20.3°, corresponding to the diffraction of the ($10\overset{-}{1}$) crystal plane of cellulose II [26]. Meanwhile, the diffraction peak of the ($10\overset{-}{1}$) crystal plane of cellulose II is seen in the curve of the warp yarn via hot air rather than in the weft yarn, indicating that under no tension or relatively low tension, some type I cellulose of weft yarns is transformed into type II cellulose, of which the thermal stability is significantly higher than that of cellulose I [27].

### 3.6. Thermal Properties

The samples for the following analysis are CF/MH with adhesion ratio of MH of 7.99/wt/%, warp and weft shrinkage of 1.09% and 15.24% via hot air, and 7.36/wt/%, 1.40%, and 2.13% via hot stamping (Figure 5). As shown in Table 3, the decomposition temperature of CF is 359.5 °C, which is lower than that of CF/MH via hot stamping (367.2 °C) and hot air (371.0 °C), indicating that the decomposition temperature of CF can be effectively increased after MH is adhered. Meanwhile, the weight loss ratio is also reduced after the MH is adhered as shown in Table 3, which indicates that the adhesion of MH improves the thermal stability of CF. Comparing with hot stamping, lower weight loss, higher char residue, and higher thermal decomposition temperature are found for CF/MH adhered via hot air due to higher adhesion ratio of MH. Moreover, the higher thermal stability of type II cellulose observed in XRD data and more contact of MH to cellulose due to MH concealing in the cellulosic film on fiber surface in SEM images possibly contribute to the above stable thermal properties.

### 3.7. Combustion Behavior

The heat release rate (HRR) is measured by a cone calorimeter as shown in Figure 6. The heat release of CF, CF/MH via hot air, and CF/MH via hot stamping increase rapidly within 0-100 s, indicating that heat release of each fabric mainly occurs in the primary and main combustion pyrolysis stages [28]. The peak heat release rate (PHRR) of CF/MH via hot air, CF/MH via hot stamping, and CF are 199 KW/m^2^, 218 KW/m^2^, and 232 KW/m^2^, implying heat absorption of immobilized MH when generating MgO during main combustion decomposition. Nevertheless, higher HRR of CF/MH after 100 s suggests pyrolysis of higher amount of char due to facilitation of char formation by MgO, which is consistent with lower weight loss in Table 3. Moreover, the total heat release per unit mass is calculated by dividing the total heat release by mass of each sample. The total heat release (THR) per unit mass of CF/MH via hot air and CF/MH via hot stamping is 1.63 MJ/(m^2^·g) and 1.86 MJ/(m^2^·g), which are decreased by 20.9% and 10.0% compared to that of CF (2.06MJ/(m^2^·g)), respectively. It can be seen that the immobilization of MH reduces the amount of heat release during the combustion of CF. In particular, CF/MH via hot air has a better effect of suppressing heat release of the fabric due to higher MH immobilization.

### 3.8. Physical Properties

It can be seen from Table 4 that the CF/MH adhered via hot stamping has a higher bending rigidity and lower activity than CF/MH via hot air. The swelling effect of IL on the cotton fiber under heat and pressure microscopically promotes the swelling of the cotton fiber and macroscopically constrains the swelling of the fabric. The resultant larger internal stress of the fiber after swelling and shrinking resulted in greater rigidity for CF/MH adhered via hot stamping. In addition, the larger warp shrinkage and weft shrinkage of CF/MH via hot air increase warp and weft density of the fabric. Therefore, the breaking strength and breaking elongation of the CF/MH via hot air are greatly increased compared with CF and CF/MH via hot stamping.

## 4. Conclusions

A simple physical adhesion method of MH has been used to improve flame retardancy of CF via spray of in situ synthesized MH/IL suspension and subsequent treatment of hot air or hot stamping. Comparing to CF, the afterflame time and afterglow time are shortened by 52.4% and 92.6% for CF/MH via hot stamping and by 42.9% and 91.4% for CF/MH via hot air, respectively. Lower weight loss of CF/MH of 79.7% and 88.5% are achieved via hot air and hot stamping in comparison with CF of 91.4%.The total heat released per unit mass of CF/MH via hot air and CF/MH via hot stamping are decreased by 20.9% and 10.0% compared to that of CF. Lower weight loss, lower heat release, higher char residue, and higher thermal decomposition temperature are found for CF/MH adhered via hot air due to higher MH adhesion and more contact of MH particles with cellulose. When the MH loading ratio exceeds 6.28/wt %, shorter afterflame time and afterflow time shorter than 5 s are more easily obtained via hot stamping than hot air. The swelling of cellulose by IL changes the crystal form of the cellulose to a certain extent under the condition of relatively small tension or without tension. Improved breaking strength and slightly decreased flexibility are obtained after adhesion of MH to CF.

## Figures and Tables

**Figure 1 polymers-12-00259-f001:**
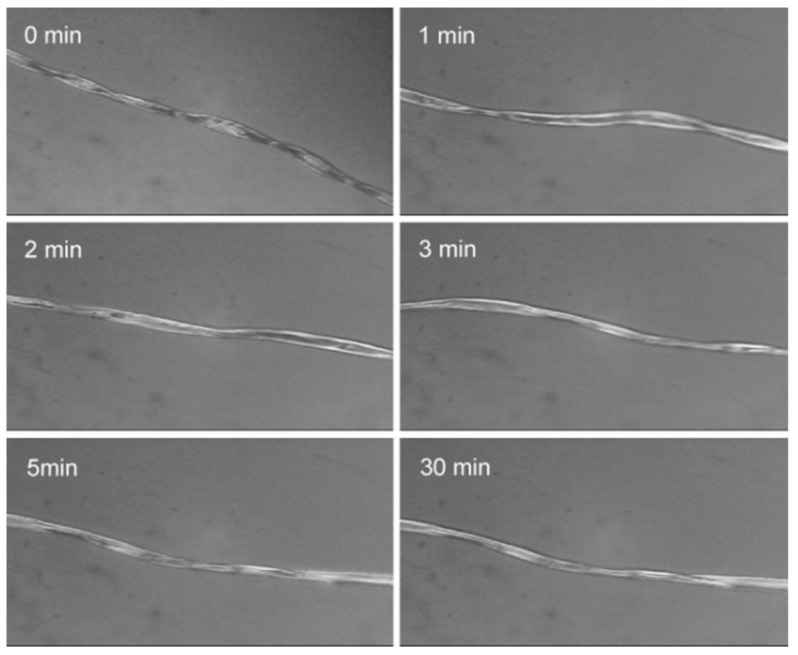
Swelling of cotton fibers by IL with time varying.

**Figure 2 polymers-12-00259-f002:**
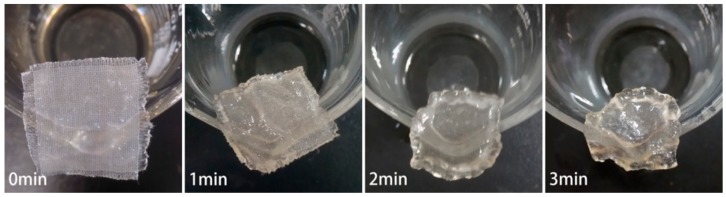
Swelling of CF in IL with time varying.

**Figure 3 polymers-12-00259-f003:**
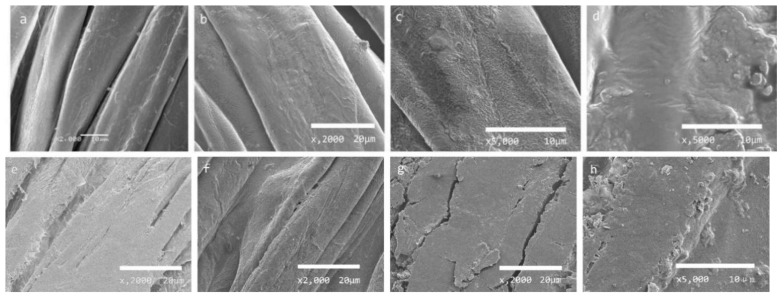
Images of CF and treated CF: (**a**) CF; (**b**) CF swelled by IL (wet state); **(c)** CF swelled by IL (wet state and high multiple); (**d**) CF swelled by MH/IL suspension (wet state); (**e**) CF swelled by IL via hot air; (**f**) CF swelled by IL via hot stamping; (**g**) CF/MH via hot air; (**h**) CF/MH via hot stamping.

**Figure 4 polymers-12-00259-f004:**
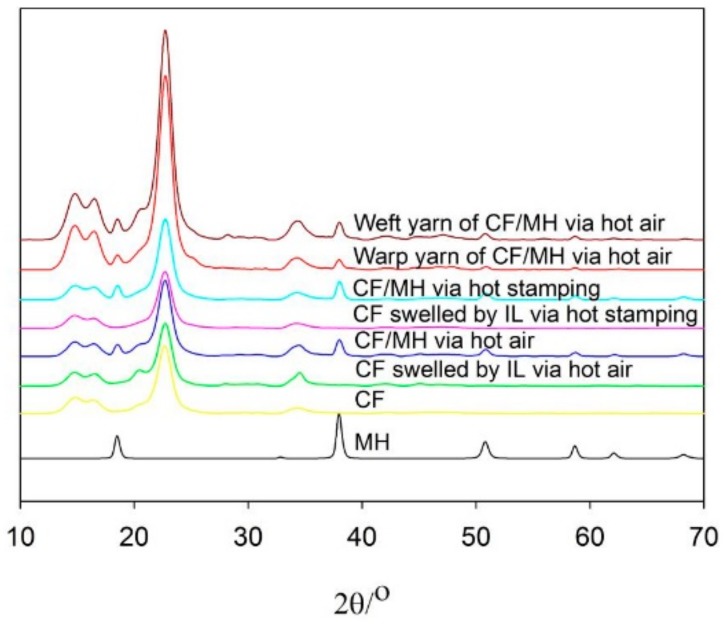
XRD curves of MH, CF, CF swelled by IL via hot air and hot stamping, CF/MH via hot air and hot stamping, warp yarn, and weft yarn of CF/MH via hot air.

**Figure 5 polymers-12-00259-f005:**
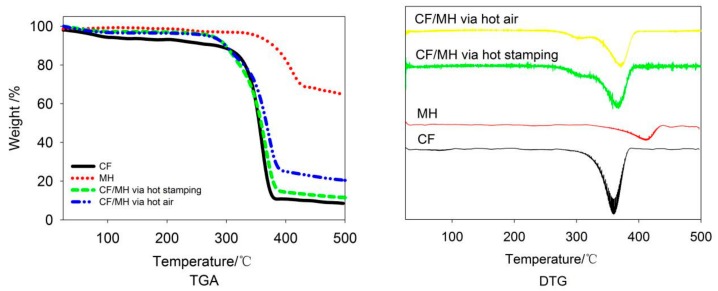
TGA and DTG curves of CF, MH, and CF/MH via hot air and CF/MH via hot stamping.

**Figure 6 polymers-12-00259-f006:**
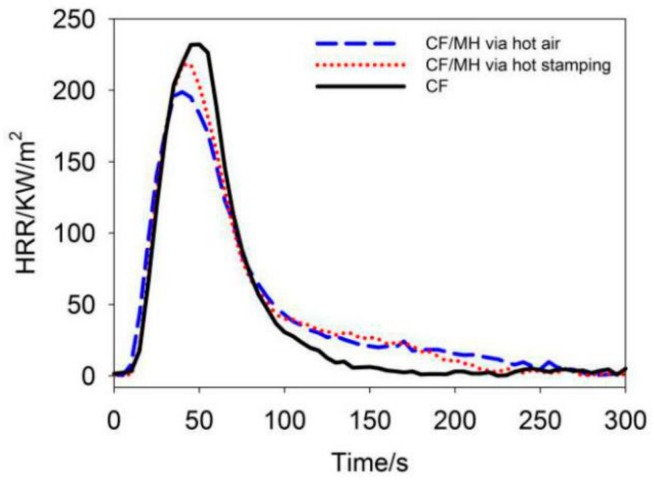
HRR curves of CF, CF/MH via hot air, and CF/MH via hot stamping.

**Table 1 polymers-12-00259-t001:** The data of flame retardancy of CF/MH under different adhesion conditions.

Spraying Temperature/Heating Time/Heating Temperature/Tension	Adhesion Ratio of MH/%	Warp Shrinkage Ratio/%	Weft Shrinkage Ratio/%	Afterflame Time/s	Afterglow Time/s
CF	0	0	0	4.2	40.8
36 °C/10 min/105 °C	4.00	9.71	38.83	8.9	16.1
39 °C/10 min/105 °C	9.89	11.43	40.95	7.1	10.6
41 °C/10 min/105 °C	9.74	12.57	42.86	11.1	10.0
39 °C/3 min/95 °C	4.20	0	4.72	1.2	15.7
39 °C/6 min/95 °C	7.99	1.09	15.24	2.4	3.5
39 °C/10 min/95 °C	9.38	0.84	31.78	6.6	2.1
39 °C/10 min/105 °C/under no tension	10.13	20.28	42.37	9.0	13.6

**Table 2 polymers-12-00259-t002:** The data of flame retardancy of CF/MH under different heating methods.

Samples		Adhesion Ratio of MH/%	Warp Shrinkage Ratio/%	Weft Shrinkage Ratio/%	Afterflame Time/s	Afterglow Time/s
90 °C	hot air	7.86	0	16.35	4.6	7.9
95 °C		9.38	0.84	31.78	6.6	2.1
105 °C		9.89	11.43	40.95	7.1	10.6
80 °C	hot stamping	6.28	0.83	1.90	3.2	4.0
90 °C		7.36	1.40	2.13	2.0	3.0
95 °C		8.02	2.00	2.86	2.5	4.6
105 °C		10.33	2.86	3.46	2.8	5.2

**Table 3 polymers-12-00259-t003:** The thermal decomposition temperature and weight loss of MH, CF, and CF/MH via hot air and hot stamping.

Sample	Weight Loss/%	Maximum Decomposition Temperature/°C
MH	35.4	411.3
CF	91.4	359.5
CF/MH via hot air	79.7	371.0
CF/MH via hot stamping	88.5	367.2

**Table 4 polymers-12-00259-t004:** Physical properties of CF and CF/MH via hot air and hot stamping.

Samples	Bending Rigidity/cN.mm^−1^	Activity Ratio/%	Breaking Strength/N	Breaking Elongation Ratio/%
CF	0.95	52.44	593	12.9
CF/MH via hot air	1.09	45.31	787	13.7
CF/MH via hot stamping	2.15	17.90	662	12.1

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
