# Peer review of "Facile Preparation of Flame Retardant Cotton Fabric via Adhesion of Mg(OH)_2_ by the Assistance of Ionic Liquid"

_polymers, 2020, doi:10.3390/polym12020259_

Round 1

Reviewer 1 Report

The authors aimed at decreasing the flammability of cotton fabrics via adhesion of MH. They used as-prepared ionic liquids to swell the fabric, in order to improve the adhesion of MH.

The idea is new; however, there are some points to be fixed prior publication.

First of all, some evidence of the swelling of the fabrics by the IL should be given. What was the starting size of the CF before spraying, and how did it change?

How were the adhesion ratios of the MH determined?

How can you explain that the FR properties are much better, if the shrinkage is small?

Starting from section 3.4, it should be given that which samples were analysed, especially when the thermal properties are discussed. As the absorbed amount of MH was very different, one can not conclude on the method if the amount of the MH is not the same.

In table 4, what is activity ratio? How it was determined?

Do the authors consider their samples as effectively flame retarded cotton fabrics?

Author Response

Point 1: First of all, some evidence of the swelling of the fabrics by the IL should be given. What was the starting size of the CF before spraying, and how did it change?

Response 1:  In Section 3.1. CF swelling in IL, the evidence of swelling of the fabrics by the IL has been given. It can be seen from Figure 1, the cotton fiber becomes brighter and thicker due to swelling by IL when heated for 1 min at 80℃. Meanwhile, natural convolution of cotton fiber gradually disappears, indistinct fiber boundary is noticed. No significant change in fiber diameter was further observed with time prolonging due to limited amount of IL. As shown in Figure 2, obvious swelling phenomenon is firstly noticed for the yarns at the edge of CF and the structure of CF is visible when the CF is heated at 80°C for 1 min in excessive IL. Nevertheless, CF shrinks largely when the time reaches 2 min. The shrinkage of CF is attributed to anisotropic swelling of cotton fibers in radical and lengthwise direction. A gel state of CF is found due to partial dissolution of cellulose by IL after 3min. The above phenomena indicate that controllable swelling of cotton fibers can be obtained when adjusting IL amount and heating temperature. Moreover, obvious swelling effect of CF by IL can be achieved with limited amount of IL for several minutes above 80°C while maintaining the shape of CF, providing a reference to design reasonable process of MH immobilization. The adhesion of MH will adhere to surface cellulose fibrils after coagulation during washing of CF in water.

   The size of the CF before spraying is 36×11 cm, which is described in section 2.3 (line 84). Fabric shrinkage in warp and weft direction occurs after MH adhesion. The warp and weft shrinkage of CF are listed in Table 1

Point 2: How were the adhesion ratios of the MH determined?

Response 2: The calculation method of MH immobilization ratio is explained in Line92-97 in Section 2.3. The adhesion ratio of MH onto CF after washing were calculated by the following formula:

Adhesion ratio of MH =(m-m0)/m0*100%

m0 and m represent the mass of the cotton fabric before and after MH adhesion.

Point 3: How can you explain that the FR properties are much better, if the shrinkage is small?

Response 3: In Section 3.2, the problem is explained that the FR properties are much better if the shrinkage is small. In Line146-148, Nevertheless, weft and warp shrinkage prolongs the afterflame time and afterglow time at a high adhesion ratio of MH, respectively, since thicker and more compact CF can lower the flame spread rate.

Point 4: Starting from section 3.4, it should be given that which samples were analysed, especially when the thermal properties are discussed. As the absorbed amount of MH was very different, one can not conclude on the method if the amount of the MH is not the same.

Response 4: The sample information for analysis has been supplemented in Line 108-110 in Section 2.4. The samples for the following analysis are CF/MH with adhesion ratio of MH of 7.99%, warp and weft shrinkage of 1.09% and 15.24% via hot air, and 7.36%, 1.40% and 2.13% via hot stamping.

Point 5: In table 4, what is activity ratio? How it was determined?

Response 5: The active ratio is an indicator of the drapability of the fabric, and the fabric with a large active rate has good drapability. The active ratio was measured using the fabric style tester (YG821L, Laizhou Electronic Instrument Co., Ltd.) according to FZ/T01054.

Point 6: Do the authors consider their samples as effectively flame retarded cotton fabrics?

Response 6: It is a new trial to immobilize MH onto cotton fabric with the assistance of IL for improvement of flame retardancy. The flame retardancy of cotton fabrics was improved after MH adhesion to a certain extent. The further study is required to obtain better flame retardancy.

Reviewer 2 Report

Results presented in this manuscript are very poor, since a very small effect of surface modification of cotton fabric with IL-Mg(OH)2 dispersion on flame resistance were observed. Presented experimental results are not convincing. No correlation exists between adhesion ratio of MH [wt.%] and afterflame time [s] and afterglow time [s] as well. In fact only 2 experi-mental results (in Table 1) and 3 results (in Table  of afterflame time are positive.

I recommend to use other methods (e.g. cone calorimetry) for evaluation of fire resistance of cotton fabrics modified with IL-Mg(OH)2 dispersion.

A word "green" must be removed from a title of the manuscript, because ionic liquids are toxic compounds !

Author Response

Point 1: No correlation exists between adhesion ratio of MH [wt.%] and afterflame time [s] and afterglow time [s].

Response 1: The relationship between adhesion ratio of MH [wt.%] and afterflame time [s] and afterglow time [s] has been explained in Section 3.2 and 3.3. Regarding MH adhesion via hot air, the afterflame time and afterglow time are reduced with the adhesion ratio of MH increasing. Nevertheless, weft shrinkage and warp shrinkage prolong the afterflame time and afterglow time at a higher adhesion ratio of MH, respectively, which is explained in line 144-148.Regarding MH adhesion via hot stamping, When the MH loading ratio exceeds 6.28/wt%, the afterflame time and the afterglow time of CF can be kept within 5s via hot stamping, which is explained in line 155-156.

Point 2: I recommend to use other methods (e.g. cone calorimetry) for evaluation of fire resistance of cotton fabrics modified with IL-Mg(OH)2 dispersion.

Response 2: Many thanks for your suggestion. The combustion behavior was analyzed by the cone calorimetry(Vouch 6810) under an external heat flux of 50kW/ m2 according to ISO 5660-1.IN section 3.7. Combustion behavior, the following explanation has been supplemented. The heat release rate(HRR) is measured by a cone calorimeter as shown in Figure 6. The heat release of CF, CF/MH via hot air and CF/MH via hot stamping increase rapidly within 0-100s, indicating that heat release of each fabric mainly occurs in the primary and main combustion pyrolysis stages. The peak heat release rate(PHRR) of CF/MH via hot air, CF/MH via hot stamping and CF are 199 KW/ m2, 218 KW/ m2 and 232KW/m2, implying heat absorption of immobilized MH when generating MgO during main combustion decomposition. Nevertheless, higher HRR of CF/MH after 100s suggests pyrolysis of higher amount of char due to facilitation of char formation by MgO, which is consistent with lower weight loss in Table 3. Moreover, the total heat release per unit mass is calculated by dividing the total heat release by mass of each sample. The total heat release(THR) per unit mass of CF/MH via hot air and CF/MH via hot stamping is 1.63MJ/( m2·g) and 1.86MJ/( m2·g), which are decreased by 20.9% and 10.0% compared to that of CF(2.06MJ/( m2·g)) ,respectively. It can be seen that the immobilization of MH reduces the amount of heat release during the combustion of CF. In particular, CF/MH via hot air has a better effect of suppressing heat release of the fabric due to higher MH immobilization.

Point 3: A word "green" must be removed from a title of the manuscript, because ionic liquids are toxic compounds !

Response 3:  A word "green" has been removed from the title of the manuscript.

Round 2

Reviewer 1 Report

Due to the new methods and new results, the quality of the manuscript has been significntly improved. There are only some typos left in the ms, which can be corrected during the processing of the article.